# Burnout: A Review of Theory and Measurement

**DOI:** 10.3390/ijerph19031780

**Published:** 2022-02-04

**Authors:** Sergio Edú-Valsania, Ana Laguía, Juan A. Moriano

**Affiliations:** 1Department of Social Sciences, Universidad Europea Miguel de Cervantes (UEMC), C/Padre Julio Chevalier, 2, 47012 Valladolid, Spain; sedu@uemc.es; 2Department of Social and Organizational Psychology, Faculty of Psychology, Universidad Nacional de Educación a Distancia (UNED), C/Juan del Rosal 10, 28040 Madrid, Spain; jamoriano@psi.uned.es

**Keywords:** job burnout, job stress, occupational health

## Abstract

A growing body of empirical evidence shows that occupational health is now more relevant than ever due to the COVID-19 pandemic. This review focuses on burnout, an occupational phenomenon that results from chronic stress in the workplace. After analyzing how burnout occurs and its different dimensions, the following aspects are discussed: (1) Description of the factors that can trigger burnout and the individual factors that have been proposed to modulate it, (2) identification of the effects that burnout generates at both individual and organizational levels, (3) presentation of the main actions that can be used to prevent and/or reduce burnout, and (4) recapitulation of the main tools that have been developed so far to measure burnout, both from a generic perspective or applied to specific occupations. Furthermore, this review summarizes the main contributions of the papers that comprise the Special Issue on “Occupational Stress and Health: Psychological Burden and Burnout”, which represent an advance in the theoretical and practical understanding of burnout.

## 1. Introduction

When work and professional environments are not well organized and managed, they can have adverse consequences for workers that, far from dignifying them, exhaust them and consume their psychological resources. Burnout has become one of the most important psychosocial occupational hazards in today’s society, generating significant costs for both individuals and organizations [1,2,3,4]. Although burnout was initially considered to be specific to professionals working in the care of people [5], later evidence has shown that this syndrome can develop among all types of professions and occupational groups [6,7]. However, burnout prevalence estimates vary considerably according to the burnout definition applied. For instance, a national study of US general surgery residents found estimates varied from 3.2% to 91.4%, with 43.2% of respondents acknowledging weekly symptoms [8].

The enormous negative impact that burnout has on the work and personal lives of workers, also affecting the economy and public health of the most affected countries, has led the World Health Organization (WHO) to include this syndrome in the 11th Revision of the International Classification of Diseases (ICD-11) as a phenomenon exclusive to the occupational context. Likewise, the need to address burnout is also justified for legal reasons, such as compliance with the European Union Framework Directive on Health and Safety (89/391/EEC).

A growing body of empirical evidence shows that occupational health is now more relevant than ever due to the COVID-19 pandemic. Particularly, the pandemic has placed considerable psychological strain on healthcare workers. Since the COVID-19 outbreak, numerous studies related to burnout have been carried out with samples of frontline healthcare workers, physicians, nurses, or pharmacists across the world [9,10,11,12]. However, the lack of a baseline level of burnout before the pandemic makes it difficult to compare changes in prevalence for the same populations. Evidence from studies of the impact of past outbreaks (e.g., SARS, influenza, or Ebola epidemics) show long-term cognitive and mental health effects (e.g., emotional distress, post-traumatic stress disorder) [13]. This evidence can be useful to design interventions for healthcare workers. These are also hard times for workers in general. Teleworking full-time due to COVID-19 has received the attention of several empirical works, which analyze job exhaustion and burnout [14,15,16,17]. Teacher burnout is also the focus of an increasing number of studies [18,19,20,21]. Additionally, working parents may experience high levels of stress in the home environment during the COVID-19 pandemic, leading to parental burnout [22,23].

This review aims to understand what burnout is and its different components, how it occurs, to identify the factors that trigger burnout and the individual factors that modulate it, to identify the effects that burnout generates at both individual and organizational levels, to understand which are the main actions that can be used to prevent and/or reduce burnout, and to present the main tools that currently exist to measure burnout.

## 2. Burnout: Definition and Development of This Construct

Overall, burnout syndrome is an individual response to chronic work stress that develops progressively and can eventually become chronic, causing health alterations [24]. From a psychological point of view, this syndrome causes damage at a cognitive, emotional, and attitudinal level, which translates into negative behavior towards work, peers, users, and the professional role itself [25]. However, it is not a personal problem, but a consequence of certain characteristics of the work activity [26].

Historically, Graham Greene was the first author to use the term burnout in his novel “A Burnt-Out Case” when describing the story of an architect who found neither meaning in his profession nor pleasure in life. Later, the term was picked up and introduced in the psychological sphere by Freudenberger [27], where he described burnout as a state of exhaustion, fatigue, and frustration due to a professional activity that fails to produce the expected expectations. Initially, this author delimited it as something exclusively related to volunteer workers in a care center where all kinds of people with mental disorders and social problems attended. Because of their occupation, these workers experienced in crescendo a loss of energy to the point of exhaustion and demotivation, as well as aggressiveness towards the service users.

Shortly thereafter, Maslach [28] introduced burnout into the scientific literature and defined it as a gradual process of fatigue, cynicism, and reduced commitment among social care professionals. Years later and after several empirical studies, Maslach and Jackson [5] reformulated the concept and elaborated a more rigorous and operational definition of burnout as a psychological syndrome characterized by emotional exhaustion, depersonalization and a reduced sense of professional efficacy that can appear in caregivers (Table 1). The turning point between the two definitions is the consideration of burnout as a syndrome, with a syndrome being understood as a picture or set of symptoms and signs that exist at the same time and clinically define a particular state distinct from others.

However, some authors have argued that these three dimensions are not completely independent. Thus, it is possible to find several explanations in the literature. The difference between them lies in which is the first dimension that appears in the face of job stress (emotional exhaustion or depersonalization). Although definitive evidence has not yet been obtained, longitudinal studies have shown that there is a causal order between the key dimensions of burnout. Thus, high levels of emotional exhaustion lead to high levels of cynicism or depersonalization [29]. Likewise, empirical studies indicate that exhaustion and depersonalization constitute the core or key dimensions of the syndrome of being burned out at work, while lack of professional fulfillment is considered as an antecedent of burnout or even a consequence [30].

Finally, although Maslach and Jackson’s [5] conceptualization of burnout remains the most widely accepted, other definitions or formulations are found in the scientific literature. For example, Salanova et al. [31] reformulate such approaches and propose an extended model of burnout composed of: (1) exhaustion (related to crises in the relationship between the person and work in general), (2) mental distance that includes both cynicism (distant attitudes towards work in general) and depersonalization (distant attitudes towards the people for and with whom one works) and (3) professional inefficacy (feeling of not doing tasks adequately and being incompetent at work).

### 2.1. Subtypes of Burnout

As an alternative to the unitary definition of burnout, Montero-Marín [24] proposes that this syndrome does not always develop in the same way and that, on the contrary, there can be three variations that depend on the dedication of workers to their work activity (Figure 1). These subtypes could also be understood as stages in which there is a progressive deterioration in the levels of worker commitment to their job and have repercussions when choosing the intervention to be applied [32]. From this theoretical perspective, burnout is considered a developing condition, with a progressive reduction in levels of engagement, and evolves from enthusiasm to apathy [24]. Burnout is proposed to typically appear with the excessive involvement characteristic of the frenetic subtype. Since it is not easy to maintain this level of activity without becoming exhausted, the worker may adopt a certain protective distance. This distancing may relieve workers from overactivity, but at the cost of the frustration that emerges in the under-challenged subtype. In the long run, this leads to a reduced perception of efficacy, giving way to passive coping strategies, typically present in the worn-out subtype. The parallelism between the evolution of the syndrome and the different subtypes raises the possibility of implementing new lines of therapeutic intervention on burnout by understanding the subtypes as a succession of stages in the development of the syndrome [24]. Indeed, empirical studies suggest a progressive deterioration from the frenetic to the under-challenged and worn-out [33]. Nevertheless, more longitudinal studies are still needed to clarify the transition from one subtype to another and the evolution of the syndrome.

The frenetic subtype is typical of work contexts with overload and workers who work intensely until exhaustion. It also tends to be more frequent in jobs with split shifts, temporary contracts and, in general, situations that force workers to be much more involved to keep the job. It is the subtype of burnout in which workers show greater dedication to work. At the motivational level, these workers show high involvement and need to obtain important achievements and it has been related to a coping style based on the attempt to solve problems actively, for which they use a high number of working hours per week or are involved in different jobs at the same time. For all these reasons, this profile is associated with high levels of burnout and a feeling of abandonment of personal life and health at work.

The under-challenged subtype is typical of monotonous and unstimulating professions, with repetitive, mechanical, and routine tasks that do not provide the necessary satisfaction to workers, who state that the work is not rewarding and is monotonous. Consequently, workers show indifference, boredom, and lack of personal development along with a desire to change jobs. This subtype of burnout is related to high levels of cynicism, due to a lack of identification with work tasks, and is associated with an escapist coping style, based on distraction or cognitive avoidance.

The worn-out subtype is characterized mainly by feelings of hopelessness and a sense of lack of control over the results of their work and recognition of the efforts invested, so that they finally opt for neglect and abandonment as a response to any difficulty. It is, therefore, the profile in which the worker shows less dedication. Thus, this type of profile is strongly associated with the perception of inefficiency and a passive style of coping with stress, based mainly on behavioral disconnection, which generates a strong sense of incompetence and makes them experience feelings of guilt.

### 2.2. Why Does Burnout Appear and How Does It Develop?

Since the appearance of the term in the scientific literature, several approaches have emerged that have attempted to answer the question of why burnout appears and how it develops. In this section, we will focus on detailing the most current and empirically supported explanatory theories of burnout considering that, instead of being antagonistic to each other, they are complementary and provide a more global view of this syndrome. Specifically, the following theories are summarized: (1) social cognitive theory; (2) social exchange theory; (3) organizational theory; (4) structural theory; (5) job demands–resources theory; (6) emotional contagion theory.

#### 2.2.1. Social Cognitive Theory

This approach is characterized by giving a central role to individual variables such as self-efficacy, self-confidence and self-concept in the development and evolution of burnout [34,35]. So, this syndrome is triggered when the worker harbors doubts about their own effectiveness, or that of their group, in achieving professional goals [36,37]. These approaches were corroborated in a study conducted in a Spanish context with 274 secondary school teachers showing that burnout occurred after the emergence of professional efficacy crises [36].

The circumstances facilitate the development of inefficacy expectations or efficacy crises are the following [38]: (1) negative experiences of failure in the past, (2) lack of reference models who have gone through a similar experience and have overcome it, (3) lack of external reinforcement for the work, (4) lack of feedback on the work completed or excessive negative criticism, and (5) difficulties at work. In this way, crises of effectiveness would lead to low professional fulfillment which, if maintained over time, would generate emotional exhaustion and then cynicism/depersonalization as a way of coping with stress (Figure 2).

#### 2.2.2. Social Exchange Theory

This theory considers that burnout occurs when the worker perceives a lack of equity between the efforts and contributions made and the results obtained in their work [39]. This lack of reciprocity, which can occur with service users, colleagues, supervisors, and organizations, consumes the emotional resources of professionals, generating an emotional exhaustion that becomes chronic. From this approach, burnout can be triggered by the significant interpersonal demands involved in dealing with clients/users that become emotionally consuming. Thus, to avoid contact with the original source of discomfort, depersonalization or cynicism is used as a stress coping strategy, which ultimately leads to low personal fulfillment (Figure 3).

#### 2.2.3. Organizational Theory

This approach considers that burnout is a consequence of organizational and work stressors (see Section 3.1) combined with inadequate individual coping strategies [40,41]. Within this theory, there are two alternative models to explain the relationships between the dimensions of burnout. According to Golembiewski et al. [41], burnout starts because of the existence of organizational stressors or risk factors, such as work overload or role ambiguity, and before which some individuals show as a coping strategy a decrease in their organizational commitment, which is very similar to cynicism and depersonalization. Subsequently, the person will experience low personal fulfillment at work and emotional exhaustion, which triggers burnout syndrome. Thus, depersonalization would be the first phase of burnout, followed by a feeling of low self-fulfillment and, finally, emotional exhaustion. The alternative proposal is that put forward by Cox et al. [40]. For these authors, the emotional exhaustion caused by enduring work stressors is the initial dimension of this syndrome. Depersonalization is considered a coping strategy in the face of emotional exhaustion and low personal fulfillment is the result.

#### 2.2.4. Demands–Resources Theory

This approach postulates that burnout occurs when there is an imbalance between the demands and resources derived from work [42]. Job demands are those job factors that require sustained physical or mental effort and are associated with certain physiological costs due to activation of the hypothalamic–pituitary–adrenal axis and psychological costs (e.g., subjective fatigue, reduced focus of attention, and redefinition of task requirements). Common work demands include work overload, emotional labor, time pressure, or interpersonal conflicts. When recovery in the face of such demands is insufficient or inadequate, a state of physical and mental exhaustion is triggered.

Work resources, on the other hand, refer to the physical, psychological, organizational, or social aspects of work that can reduce the demands of work and the associated physiological and psychological costs and that can be decisive in achieving work objectives. Resources at work can be organizational in nature, but also personal (Table 2). When demands exceed resources, fatigue occurs; if this imbalance is maintained over time, fatigue becomes chronic and, finally, burnout appears. Therefore, job demands have a direct and positive relationship with burnout, especially emotional exhaustion, while the existence of job resources inversely influences depersonalization by minimizing or reducing its use as a coping strategy.

#### 2.2.5. Structural Theory

This approach maintains that burnout is a response to chronic job stress that appears when the coping strategies employed by the individual to manage job stressors fail. Initially, work stress will elicit a series of coping strategies. When the coping strategies initially employed are not successful, they lead to professional failure and to the development of feelings of low personal fulfillment at work and emotional exhaustion. Faced with these feelings, the subject develops depersonalization attitudes as a new form of coping. (The sequence is illustrated in Figure 4.) In turn, burnout will have adverse consequences both for the health of individuals and for organizations. This model has been empirically contrasted with different professional groups such as teachers or nurses [43].

#### 2.2.6. Theory of Emotional Contagion

Emotional contagion refers to the tendency to automatically imitate and synchronize facial expressions, vocalizations, postures, and movements with those of other people and, consequently, to converge emotionally with them [44]. When people work together, it is common for them to share situations and experience collective emotions, such as sadness, fear, or exhaustion. Therefore, from this theory it is considered that burnout occurs in work groups, since there are shared beliefs and emotions that are developed throughout social interaction [38]. This burnout contagion has been evidenced especially in teaching and health personnel [45], as well as between spouses (outside work). Thus, emotional contagion influences the development of burnout both inside and outside the workplace [26,46].

## 3. What Circumstances Trigger Burnout?

The antecedents are those aspects that are going to propitiate, trigger, and/or maintain people suffering from burnout syndrome. In general, these aspects can be classified into two broad categories: (1) organizational factors such as, for example, the workload or the emotional demands involved, and (2) individual factors such as, for example, the worker’s personality or coping strategies. It is important to emphasize that this syndrome is primarily a consequence of exposure to certain working conditions and not an individual characteristic such as a personality trait. Strictly speaking, therefore, the triggers of burnout would be factors related to the work (be it content, structure or relationships with users, clients, bosses, and/or colleagues). However, it is considered that, although organizational factors are capable per se of generating burnout, certain individual factors would act as moderating variables. Thus, personal aspects such as, for example, a lack of self-confidence or the use of stress-avoidance coping mechanisms could play a role in enhancing situational factors. On the other hand, other individual characteristics, such as optimism or active coping, can lessen or even slow down the negative effect of organizational factors on burnout and its consequences.

### 3.1. Organizational Factors

Regarding situational factors, reviews of the scientific literature [47] show that, in general, both the type of tasks, the way they are organized and the relationships between colleagues, bosses, and/or clients are potential burnout triggers or risk factors.

#### 3.1.1. Work Overload

Workload, both quantitative and qualitative, when excessive, requires sustained effort, generating physiological and psychological costs. Such symptoms can trigger the experience of burnout and psychological distancing from work as a self-defense mechanism [48].

#### 3.1.2. Emotional Labor

Emotional labor is understood as the psychological process necessary to self-regulate one’s emotions and show those emotions desired by the organization. It involves controlling or hiding negative emotions such as anger, irritation or discomfort to comply with the rules or requirements of the organization and objectives of the job, as well as the display of emotions not felt, such as sympathy towards customers or users, although the opposite is really felt, or tranquility in situations in which what is really felt is fear. Emotional labor will therefore involve a greater workload. In this sense, several studies have shown positive relationships between emotional labor and burnout in different professions, such as teachers [49] and HR department workers [50].

#### 3.1.3. Lack of Autonomy and Influence at Work

Lack of freedom at work when performing tasks, as well as the inability to influence decisions that affect work has been positively associated with higher levels of burnout. Conversely, when workers experience autonomy and control over their work, there are lower rates of burnout and higher rates of professional fulfillment [48]. In this line, several investigations have found negative relationships between burnout and empowerment, so that the greater the empowerment perceived by workers, the lower the levels of burnout experienced [51,52].

#### 3.1.4. Ambiguity and Role Conflict

When the worker does not know what is expected of them and/or does not have enough information about their mission (role ambiguity) or in their case the different tasks and demands to be fulfilled are incongruent or incompatible with each other (role conflict), burnout levels are increased [53].

#### 3.1.5. Inadequate Supervision and Perception of Injustice

The perception of inadequate supervision (e.g., excessively directive, and unfair by only focusing on the negative aspects without valuing achievements and efforts, or at the other extreme not at all directive or non-existent) increases the risk of developing burnout. On the contrary, a fair treatment with employees favors the increase in available resources, exerting a negative effect on emotional exhaustion in such a way that workers are less likely to develop burnout symptomatology [54].

#### 3.1.6. Lack of Perceived Social Support

Lack of social support at work, either from co-workers or supervisors, as well as internal conflicts between co-workers are considered important triggers of burnout. On the contrary, social support has been found to act as a brake on this syndrome [55].

#### 3.1.7. Poor Working Hours

The working hours conditions that make it difficult to reconcile family and professional life are another important trigger of burnout. For instance, shift work, high rotations, night work, long working hours, or a large amount of overtime are powerful triggers of burnout. Additionally, such hourly characteristics are positively related to sleep disorders, heart problems, health complaints, job dissatisfaction, decreased attention and performance, as well as an increased risk of accidents [48].

### 3.2. Individual Factors Modulating Burnout

Regarding individual factors, both personality traits and sociodemographic variables and coping strategies have been analyzed as predisposing or facilitating the development of burnout in the case of the presence of some of the organizational factors explained above. Table 3 summarizes these factors and their modulating effect on burnout: positive (they amplify the effect of social factors) or negative (they reduce the effect of social factors).

Personality influences how people perceive their work environment and, therefore, how they manage and cope with work demands and resources. Several studies [56,57,58] conclude that the personality traits posited in the Big Five model (extraversion, neuroticism, agreeableness, conscientiousness, and openness to experience; [59]) are significantly but differentially associated with burnout. Thus, it has been found that there is a negative correlation between extraversion and the components of burnout. Thus, extraversion will be a protective factor against burnout. As for neuroticism or emotional instability, positive correlations have been found with burnout. Therefore, people with less emotional stability will be more likely to suffer from burnout. Agreeableness is another personality factor that has shown a protective effect on burnout, so that more-agreeable workers tend to experience less burnout than their less-agreeable colleagues. Likewise, conscientiousness, or the tendency to behave responsibly and persistently, reduces the likelihood of burnout. Finally, openness to experience that represents aspects related to breadth of interests and creativity also has protective effects on burnout as it is positively associated with professional efficacy and negatively associated with depersonalization.

Other individual characteristics that influence the development of burnout are the external locus of control, the type A behavior pattern and having high expectations. Locus of control [60] refers to the degree to which people believe they have control over events and their lives (internal locus of control) and the degree to which they believe that events occur due to external causes such as chance or the decisions of others (external locus of control). The greater the external locus of control, the greater the probability of developing burnout, especially in ambiguous or novel situations, in which the persons believe they have little or no possibility of controllability. Type A behavior pattern is characterized by competitiveness, impulsivity, impatience, and aggressiveness, and has been widely implicated as a health risk factor. This behavior pattern is positively related to the emotional exhaustion and depersonalization factors of burnout. Finally, the expectations that employees have regarding their work are related to the level of burnout, such that higher expectations and higher goal setting lead to greater efforts and thus higher levels of emotional exhaustion and depersonalization [47,48]. The person’s level of involvement also seems to be important. Specifically, over-involvement has also been proposed as a potent trigger, especially when it may be impossible to achieve goals. This mismatch between expectations and realities can lead to frustration and burnout in workers.

In terms of sociodemographic variables, reviews of studies [47,48] point to an inverse relationship between age and burnout, such that people will experience lower levels of burnout as their age increases. However, the results are not always so consistent. A systematic review of the determinants of burnout [61] found a significant relationship between increasing age and increased risk of depersonalization, although on the other hand there is also a greater sense of personal accomplishment. Regarding gender, most studies indicate that emotional exhaustion and low professional fulfillment tend to be more common among women while depersonalization is more frequent in men. In relation to marital status, workers who are single (especially men) seem to be more exposed to burnout compared to those who live with a partner. However, such findings seem to be more appropriate in men, as in the case of working women, it constitutes an additional risk factor since working women are usually responsible for household chores and, therefore, this may pose a difficulty in reconciling personal and professional life.

Coping strategies are another variable that play an important role in the development of burnout [62,63]. Although there are several classifications of coping strategies, the most established one is the distinction between problem-focused coping and emotion-focused coping [64]. Problem-focused coping represents an attempt to act directly on the stressful situation, whereas emotion-focused coping focuses on modifying negative emotional responses to stressful events, avoiding intervening on them. Empirical evidence suggests that, in general, avoidance and emotion-focused coping are positively related to burnout, that is, they favor it, whereas active and problem-focused coping are negatively related to burnout, that is, they reduce it. However, not all emotion-focused coping strategies increase burnout, as social support-seeking, reappraisal, and religious support, in some cases, have protective effects on burnout [55]. On the other hand, it has also been proposed that the effectiveness of problem-focused coping may depend on the control that individuals can exert over potential stressors in the work environment. Specifically, the use of problem-focused active coping strategies when there is little possibility of controlling and/or changing environmental stressors may exacerbate the undesirable effects of work stress; in such situations it is more advisable to employ coping strategies to facilitate adaptation to the situation. Therefore, one cannot be blunt in concluding that emotion-focused coping strategies are always negative since problem-focused coping only seems adaptive in controllable situations, while avoidance-oriented coping is adaptive in situations that are difficult to control [65].

### 3.3. Future Research

This section has focused on summarizing the main triggers of burnout. However, since burnout symptoms develop and evolve differently depending on individual characteristics (e.g., personality or coping strategies) and the work environment (e.g., job demands or leadership styles), it is necessary to continue advancing the knowledge of which are the personal factors that in combination with certain contextual triggers produce greater or lesser symptomatology. For example, when faced with the same stressor, do all personality types experience the same symptoms and consequences? Which personalities are more vulnerable to developing burnout when faced with specific triggers? Which are the most potentially harmful combinations of individual characteristics and contextual triggers? And which are the least? From a temporal perspective, it would also be necessary to carry out more longitudinal studies to study the evolution of symptomatology.

Finally, and because of the increase in home working during the COVID-19 pandemic, it would also be interesting to examine whether teleworking may cause a greater or lesser occurrence of this symptomatology, compared to face-to-face work, as well as to examine possible differences depending on the sector of activity.

## 4. Consequences of Burnout

Burnout results in a series of adverse consequences both for the individuals who suffer from it and for the organizations in which these professionals work. These consequences are initially of a psychological nature, but maintained over time, they translate into adverse effects on the physical/biological health and behaviors of workers, which in turn will have undesirable organizational consequences [66].

### 4.1. Psychological Consequences

The psychological alterations generated by the syndrome of being burned out at work occur at both cognitive and emotional levels. Different studies have associated this syndrome with concentration and memory problems, difficulty in making decisions, reduced coping capacity, anxiety, depression, dissatisfaction with life, low self-esteem, insomnia, irritability and increased alcohol and tobacco consumption [66,67]. Other researchers have also shown that this syndrome can pose a significant risk of suicide [68].

### 4.2. Health Consequences

Several reviews of studies conclude that employees with higher levels of burnout are more likely to suffer from a variety of physical health problems such as musculoskeletal pain, gastric alterations, cardiovascular disorders, headaches, increased vulnerability to infections, as well as insomnia and chronic fatigue [69]. Burnout has also been found to dangerously increase blood cortisol levels [70] and constitutes an independent risk factor for type 2 diabetes [71]. Now, the way these symptoms manifest themselves is not the same in all individuals, nor do they all have to occur.

### 4.3. Behavioral Consequences

In addition to physical and psychological health problems, in general, burnout is also directly related to job dissatisfaction [72], low organizational commitment [66], increased absenteeism [73], turnover intention [74], and reductions in performance [47]. On the other hand, some employees with burnout syndrome may justifiably leave their job; however, others decide to remain working [75]. This may lead to work presenteeism (i.e., individuals go to work, although they do not really fulfill their responsibilities due to health issues). In addition, burnout can lead to deviant and counterproductive behaviors in workers, aggressiveness among colleagues and towards users, alcohol and psychotropic drug use, misuse of corporate material, or even theft [68,69,75,76].

However, the form and evolution of these individual consequences (psychological, health, and behavioral) is not the same in all cases. In this sense, and although it is not always easy to delimit them, four levels of burnout syndrome have been described [77]:
Mild: those affected have mild, unspecific physical symptoms (headaches, back pain, low back pain), show some fatigue, and become less operative.Moderate: insomnia, attention and concentration deficits appear. At this level, detachment, irritability, cynicism, fatigue, boredom, progressive loss of motivation, making the individual emotionally exhausted with feelings of frustration, incompetence, guilt, and negative self-esteem.Severe: increased absenteeism, task aversion and depersonalization, as well as alcohol and psychotropic drug abuse.Extreme: extreme behaviors of isolation, aggressiveness, existential crisis, chronic depression, and suicide attempts.

### 4.4. Organizational Consequences

The negative consequences experienced at the individual level by workers with burnout translate into low motivation and performance that can extend to the work unit and the organization, causing a reduction in the quality of services [78]. Likewise, employees suffering from burnout influence the rest of the organization, causing greater conflicts or interrupting work tasks, thus reducing production and increasing production times [67]. Therefore, as indicated in the emotional contagion theory, burnout can cause a “contagion effect”, generating a bad working environment [45]. This syndrome also usually generates significant economic losses as a consequence of absenteeism, loss of efficiency and counterproductive behaviors [76].

### 4.5. Future Research

It would be interesting to examine in depth the relationships between the psychological alterations caused by burnout and the effects on workers’ health, safety, and performance. For example, how psychological damage caused by burnout influences workers’ attitudes and behavior, and exploration of the possible modulating role of individual factors and certain organizational characteristics (i.e., leadership, organizational climate, cohesion among workers). In addition, longitudinal studies would be necessary to analyze the possible relationship between the different consequences of burnout and productivity.

## 5. Prevention Strategies

Now we have established what burnout is and what circumstances trigger it, in this section we will focus on how to act both to avoid and to reverse its occurrence and consequences. First, the most appropriate type of preventive intervention should be selected. Primary prevention is aimed at all workers and its purpose is to reduce or eliminate organizational risk factors to prevent the occurrence of burnout. Primary prevention is the most consistent with the principles of an occupational risk prevention management system by providing workers with adequate support, job adaptations, information, and adequate training to deal with this psychosocial risk.

Secondary prevention, on the other hand, is carried out once the first symptoms of burnout have appeared, so it is not aimed at all workers, but only at those who are already affected and its purpose in general is that such symptoms do not evolve further, improving the way in which the person responds to these stressors. These interventions are aimed more at individuals than at the organization, bringing about changes in attitudes and improving their coping resources, which does not imply that there are no organizational interventions as well. Finally, tertiary prevention focuses on employees who are already burned out at work. The aim of this type of prevention is to reduce the most severe harms (e.g., serious health problems and/or poor job performance). Since this type of intervention is aimed at trying to resolve the damage to the worker’s physical and/or psychological health, it is considered reactive and not strictly speaking prevention, but treatment.

From another perspective, we will classify the interventions considering the promoter of the intervention, that is, who organizes, decides and, if necessary, finances the actions to be carried out. In this sense, interventions can be classified as follows: (1) promoted by the organization, which in turn could be subdivided into actions directed at the organizational and job structure and actions directed at employees, and (2) promoted by individuals, which could also be subdivided into interventions directed at oneself as an individual and interventions directed at improving one’s interaction with the organization and with aspects of the job (Table 4).

### 5.1. Organizational Interventions Aimed at Work Structure

The following is a description of interventions that generally focus on reducing work stressors and increasing the organizational resources available to workers [79,80].
Improving job characteristics. These actions are mainly aimed at quantitative workload reduction and qualitative work improvement through two main strategies:
(1)Work redesign. This measure aims to partially change the objectives and tasks of the job while improving the quality of work by eliminating structural and/or procedural elements that interfere and generate stress [81]. It could also be considered job redesign the enrichment of jobs through the incorporation of new and more stimulating tasks that make the job more motivating and rewarding.(2)Modification of exposure times to potential stressors. This can be completed by reducing the time in which the worker is exposed to the most stressful elements of the job (such as, for example, attention to users or patients) through job rotation, or, if necessary, by performing other tasks or activities [82,83].Humanization of schedules and implementation of work–life balance plans. This intervention involves organizing and making work schedules and shifts more flexible to allow for the reconciliation of personal and professional life [84]. In this sense, variable work shifts and long working hours exceeding 8 h should be eliminated.Managers’ leadership development. Supervisor support and leadership is considered an important work resource capable of reducing burnout levels in employees. However, not all supervisors employ an adequate leadership style. In this sense, several studies have shown that authentic [54], transformational [85], and servant [86] leadership styles are related to decreased burnout and have positive effects on employees’ psychological resources [87]. For this reason, these are the leadership styles that should be developed and trained to avoid the occurrence of this syndrome. Additionally, the performance of leaders and specifically leadership behaviors should also be regularly evaluated by the individuals working with them to identify potentially adverse aspects that could trigger burnout.Use of rewards and incentives that are not only financial. Employees can be motivated by rewards that do not always need to be of a financial nature. Recognizing work well done is a very efficient way to increase workers’ motivation levels and prevent burnout [48]. As indicated previously, one of the factors causing efficiency crises, which in turn were triggers of burnout, was the lack of reinforcement and appraisal by supervisors. In addition to recognition of accomplishment, other types of rewards such as greater time flexibility (which can facilitate work–life balance) or protected time to achieve personally meaningful work goals can enhance well-being. In contrast, employing simple financial rewards may be less effective by encouraging overwork and pressure to achieve goals, which promote burnout.Development of welcoming programs. Since role conflicts and ambiguities are potential triggers of burnout, it is advisable for organizations to develop welcoming processes for new workers, where the mission of the position, tasks, and objectives to be fulfilled are explained with absolute clarity and they are progressively introduced to the most stressful elements of the job, always offering support from the supervisor or other colleagues [88].Burnout monitoring and design of customized plans. This consists of periodically conducting surveys and measurements of workers to “monitor” the organization’s burnout levels and compare the scores of workers according to units, location, position, supervisor, etc. (e.g., [89]). The aim is basically to identify the appearance of the first symptoms, thus preventing the syndrome from becoming chronic. It is important that, in addition to the levels of burnout, the organization identifies as precisely as possible the risk factors in the work environment that may be present to eliminate or minimize them. Additionally, since the specific way in which symptoms manifest themselves and which dimension is dominant varies in each work unit, to be effective it will be necessary to design interventions specific to the causes and consequences/symptoms identified.Institutionalization of occupational health and safety. This intervention refers to the obligation of organizations to incorporate in their structure departments or devices in the form of agreements with other entities to ensure the health and reduction of burnout in workers [90]. This intervention translates into (e.g., [91]):
(1)Delivery of psychoeducational workshops on stress and burnout that can be scheduled in the same organization or by outsourcing the service.(2)Counseling services for workers with work-related problems. This action can be carried out within the organization or by outsourcing the service by referring the employee to a counseling specialist.(3)Referral to specialized health promotion services such as psychologists and medical specialists.

### 5.2. Interventions Promoted by the Organization Aimed at Employees

This type of intervention basically aims to increase the personal resources of employees to manage stressors at work, which in turn helps to reduce burnout levels.
Training. Through training, employees can acquire new skills and technical knowledge that increase their coping resources and improve their self-efficacy expectations. However, in addition to technical skills related to the job, organizations should plan training actions aimed at developing other types of personal and social skills that facilitate workers to implement individual strategies to promote their well-being and adjustment to the job [88,92]. Table 5 includes examples of training actions to prevent or manage burnout.Strengths-based interventions. Strengths-based interventions work from the premise that people have personal resources that can be used to cope with adversity. Using strengths is intrinsically motivating and satisfying. A strengths intervention typically unfolds in three phases, as described in Table 6.Coaching and guidance. These are non-directive methods that encourage employees to regain control of their emotional state and well-being on their own, so the coach/counselor will not “prescribe” any treatment. Instead, the coach/counselor will guide the employee to come up with (or with some assistance) coping strategies on their own [93]. This type of intervention is usually typical of secondary prevention, in the early stages of the syndrome, when it is assumed that the person still has the capacity to redirect it.Creation of support groups. Peer and team support has always been critical in helping professionals cope with the difficulties and challenges of day-to-day life. This support encompasses a wide range of activities, including the celebration of achievements or the creation of formal support groups. In this sense, organizations should incorporate activities into work processes that are conducive to such a sense of community as dedicating time to share ideas and knowledge about how to act and deal with day-to-day professional challenges [88]. Support groups refer to any group of coworkers, whether formal (expressly created by the organization) or informal (not created by the organization but arising spontaneously) that meet regularly to exchange information, give each other emotional support and/or solve work problems. What these groups have in common is that they offer recognition for work completed (even if objectives have not been achieved), comfort, help, and companionship. The primary objective of the support groups is to reduce the professionals’ feelings of loneliness and emotional exhaustion, as well as the exchange of knowledge to develop effective ways of dealing with problems. This intervention (e.g., two hours every two weeks) is one of the most widely employed interventions for intervening on burnout and its benefits have been repeatedly demonstrated [73]. While the creation of support groups is an individual focus intervention, in many cases it is encouraged by the organization, or should be.

### 5.3. Individual-Focused Interventions Promoted by the Individual

These types of actions are initiated and determined by the workers themselves and are aimed at improving their emotional and physical state completely outside the work environment, including physical exercise, mindfulness, self-assessment and, where appropriate, psychotherapy.
Physical exercise. Several studies have shown the positive effect of physical activity as a moderating variable of the effects of burnout on the health of workers [94,95]. Physical exercise can be used in primary, secondary and, where appropriate, tertiary prevention.Mindfulness training. A systematic review [96] of various specialized databases published between 2008 and 2017 concluded that mindfulness practice is effective in reducing burnout syndrome, both in its total values and in those corresponding to its dimensions, mitigating the negative psychosomatic and emotional effects of the syndrome, and increasing other positive ones such as empathy or concentration.Self-assessment. This intervention involves the self-observation of possible signs that could point to burnout. The way to do this is, for example, by keeping a diary of stress symptoms and related events such as specific symptoms, thoughts, feelings, and ways of coping with them. On the other hand, in addition to this type of diary, it is also important to measure the degree of burnout with a properly validated test, such as those indicated in the following section, and to compare one’s own score with that of a reference group or with oneself over time.Psychotherapy. Psychotherapeutic treatment of burnout syndrome is carried out in the most severe and serious cases (i.e., when the syndrome and its consequences are already being suffered). Psychotherapeutic treatment basically consists of developing emotional self-regulation and relaxation skills, problem solving, development of self-efficacy, and assertiveness and is generally based on the principles of cognitive-behavioral therapy [73,92]. This intervention may be funded by the organization; however, it will always be the individual him/herself who will make the decision to initiate a psychotherapy process. There are three types of techniques used to reduce burnout:
(1)Cognitive techniques: these are aimed at the individual reevaluating and restructuring their appreciation and vision of stressful or problematic situations, so that they can deal with these situations more effectively. This type of technique is useful because people perceive situations subjectively and individually and, therefore, in a biased way. Cognitive techniques are aimed at identifying and modifying errors in the perception of reality to influence the emotions they provoke and the behavior they trigger.(2)Physiological deactivation techniques: the aim of this type of technique is to teach the person mechanisms to control, through relaxation, the increased physiological activation and anxiety caused by stressful stimuli.(3)Training in healthy lifestyle habits: physical exercise, a balanced diet, and restful sleep can help to reduce the symptoms of burnout.

### 5.4. Individually Driven, Work-Focused Interventions

These interventions are also initiated and determined by workers, but in this case, they are aimed at improving the work environment.
Time management. Employees who are at risk of burnout often feel that they lack the time to fulfill all their responsibilities or that they work long hours with no time for personal use and rest. Self-management of time consists of correctly planning one’s time by making efficient use of the time available, organizing tasks realistically, and delegating them when appropriate, as well as dedicating daily time for personal activities and recreation [79,80]. Although this intervention is promoted by each worker, to facilitate proper time management, organizations as indicated above can or should provide training and coaching actions to their workers [97].Job crafting. Unlike job redesign (explained above), which is managed and planned by the organization, job crafting is an individual bottom-up intervention, initiated by the employees themselves, which consists of actively modifying their job (as long as the job mission is fulfilled) by reconfiguring the way they approach tasks and negotiating the job content, allowing employees to adjust their jobs to their personal knowledge, skills and abilities, and to their preferences and needs. In other words, through job crafting, the work to be performed does not change but is adjusted to experience it in a more meaningful way. These adjustments can be of four types [98] and are summarized in Table 7.

### 5.5. Future Research

Evaluation research on the success or failure of intervention strategies aimed at preventing or containing burnout is stilled needed. The interventions presented in this section offer a general and broad view of how to deal with burnout. However, since this syndrome depends on and develops idiosyncratically according to personal factors as well as working conditions, future lines of research should focus on analyzing which are the most efficient interventions according to individual characteristics and situational triggers. In addition, it would be optimal to establish comparisons between different interventions aimed at both the individual and the organization level. Furthermore, it is necessary to analyze the possible interaction between interventions and whether the combination of several of them is potentiating, inhibiting, or redundant. Finally, it would also be interesting to establish longitudinal studies to detect which of these interventions are more effective in the long term.

## 6. Assessment and Measurement

When it comes to assessing burnout, several tools (scales and questionnaires) have been developed and validated in different countries. These tools can be classified into two broad categories: (1) generic instruments (i.e., instruments aimed at assessing the syndrome, without differentiating by professional occupations; the main difference between these instruments is the burnout theoretical model they consider and what other aspects, if any, they evaluate), and (2) specific instruments aimed at evaluating burnout in specific occupations (e.g., nurses, psychologists, physicians) or even out of job (e.g., sports, school and parental relationships). Table 8 shows the main instruments currently available for assessing burnout.

### 6.1. Generic Instruments

Maslach Burnout Inventory (MBI; [5]). The most widely used and validated tool for measuring burnout. At first, this tool was designed exclusively to measure burnout in personnel in the care sector and was called the Maslach Burnout Inventory-Human Services Survey (MBI-HSS). However, research and epidemiological studies showed that burnout can occur in any occupation and sector of activity, and for these reasons Schaufeli et al. [99] developed the definitive tool, the MBI-GS (Maslach Burnout Inventory-General Survey), based on the previous one and applicable to all occupations and jobs. This instrument has 16 items distributed in three dimensions: emotional exhaustion, cynicism, and reduced professional fulfillment. Thus, high scores on these dimensions would be indicative of burnout. This tool has subsequently been validated in different cultural and work contexts, such as Spanish [6], Italian [100], French [101], Chinese [102], and Arabic [103], among others.

Questionnaire for the Evaluation of Burnout Syndrome (CESQT; [104]). The CESQT consists of twenty items that are grouped into four dimensions: (1) enthusiasm for work: this is defined as the individual’s desire to achieve work goals because it is a source of personal pleasure. Low scores in this dimension indicate high levels of burnout; (2) psychic burnout: this is defined as the occurrence of emotional and physical exhaustion because of work; (3) indolence or the presence of negative attitudes of indifference and cynicism towards the organization’s customers; and (4) guilt: this is defined as the appearance of feelings of guilt for the behavior and negative attitudes developed at work, especially towards people with whom work relationships are established. This instrument has two different versions: the main version (CESQT), which is applied to workers who work with people (e.g., psychologists, teachers, or doctors) and the “Professional Disenchantment” version (CESQTDP), which is administered to those workers who do not work in direct contact with people. Although this tool was originally designed in a Spanish context, throughout these years the CESQT has also had a great reception and a wide development in different countries. It has been translated, adapted and validated in Germany [105], France [106], Italy [107], Portugal [108], and Poland [109]. In Anglo-Saxon literature, the use of the CESQT is regularly cited as the Spanish Burnout Inventory (SBI; e.g., [110,111]), and alludes to the theoretical model from which it starts, highlighting that among its strengths is the fact of collecting a broader vision of burnout than other instruments by including the dimension of guilt [67]. The wide dissemination of the instrument and its quality as a psychological assessment tool has favored the American Psychological Association (APA) to include it in its database of psychological tests.

Copenhagen Burnout Inventory (CBI; [112]). This scale allows the assessment of context-free burnout. It is composed of three main factors: (1) personal burnout, (2) work-related burnout, and (3) client-related burnout.

Oldenburg Burnout Inventory [113]. This inventory was developed to measure burnout across various occupational groups and measures two dimensions of burnout: (1) exhaustion, which is the primary symptom of burnout, and (2) disengagement from work.

Burnout Clinical Subtypes Questionnaire (BCSQ; [114,115]). The questionnaire consists of 36 items and measures the different properties of each clinical subtype. Each subtype consists of several facets: involvement, ambition, and overload of the frenetic type; indifference, lack of development, and boredom of the under-challenged type; and finally, neglect, lack of acknowledgement, and lack of control of the worn-out type. This questionnaire was originally developed in Spain, but recently it has been validated for other cultures such as Latvia [116] and Germany [117]. In its short version (BCSQ-12), consisting of 12 items, only one subscale of each subtype is analyzed (i.e., overload, lack of development, and neglect).

Burnout Assessment Tool (BAT; [118]). This tool is based on an alternative, comprehensive conceptualization of burnout, and includes all relevant elements that are associated with burnout. The questionnaire contains 33 items and consists of the BAT-C and BAT-S. The BAT-C assesses the four core dimensions: (1) exhaustion, (2) cognitive, (3) emotional impairment, and (4) mental distance). The BAT-S assesses two atypical secondary dimensions that often co-occur with the core symptoms: (1) psychological complaints, and (2) psychosomatic complaints.

Shirom–Melamed Burnout Questionnaire (SMBQ; [119]). The instrument comprises 22 items which consists of the following sub-scales: (1) emotional exhaustion, (2) physical fatigue, (3) cognitive weariness, (4) tension, and (5) listlessness. Later development of the instrument resulted in the Shirom–Melamed Burnout Measure (SMBM; [120]), which included 14 item divided in three subscales; (1) physical fatigue, (2) emotional exhaustion, and (3) cognitive weariness.

### 6.2. Specific Instruments

Maslach Burnout Inventory-Human Services Survey (MBI-HSS; [5]). This is a 22-item survey, applicable to human services jobs, for instance, clergy, police, therapists, social workers, medical professionals. The MBI-HSS (MP), adapted for medical personnel, and MBI-Educators Survey (MBI-ES), adapted for educators, are available online at https://www.mindgarden.com/117-maslach-burnout-inventory-mbi (accessed on 26 December 2022).

Brief Burnout Questionnaire Revised for nursing staff [121]. This instrument is an alternative tool to the MBI-HSS (MP). The questionnaire comprises 21 items that evaluate not only the syndrome itself, but also its antecedents and consequences. These items are gathered into four factors: (1) job dissatisfaction, comprising four items; (2) social climate, made up of three items; (3) personal impact, made up of four items, and (4) motivational exhaustion, comprising four items.

Physician Burnout Questionnaire-PhBQ [122]. This is another alternative instrument to the MBI-HSS (MP). The PhBQ contains 17 items and includes four subscales: burnout syndrome (PhBSS), antecedents (PhBAS), consequences (PhBCS), and personal resources (PPRS).

Teacher Burnout Questionnaire [123]. This questionnaire examines the burnout of teachers and is based on Maslach, Jackson and Leiter’s original instrument ([28]). The questionnaire comprises 14 items.

Psychologist’s Burnout Inventory—PBI [124]. This instrument measures four factors related to burnout among psychologist: control (three items assessing control over work activities, schedule, and decisions), overinvolvement (three items assessing feelings of responsibility for and spending time thinking about or dealing with clients), support (three items assessing emotional and instrumental support from colleagues), and negative client behaviors (six items assessing the experience of aggressive, dangerous, or threatening client behaviors). A revision of this instrument (PBI-R) was developed by Rupert et al. [125].

Athlete Burnout Questionnaire [126,127]. This tool is adapted to sport environments, and it is composed of 15 items organized in three dimensions: emotional/physical exhaustion, reduced sense of accomplishment and devaluation.

School Burnout Inventory-SBI [128]. This inventory comprises nine items grouped in three dimensions: (a) exhaustion at school, (b) cynicism toward the meaning of school, and (c) sense of inadequacy at school.

Parental Burnout Inventory [129]. This instrument assesses parental burnout syndrome, including exhaustion, distancing, and inefficacy.

### 6.3. Future Research

The main objection that could be made to the questionnaires presented above is that they are self-reported measures that focus especially on quantifying the burnout factors (emotional exhaustion, cynicism, and professional efficacy). However, since the burnout phenomenon is complex, more tools should be designed that consider both the antecedents and the physical and psychological consequences of burnout, thus offering a more global vision of this syndrome. As noted by Shirom [130], burnout measures should be analyzed within the framework of theoretical models that also consider causes and effects of burnout, as well as correlates. This type of instrument would, in turn, allow the development of more individualized and personalized interventions and treatments.

Moreover, different theoretical conceptualizations of burnout have led to the proliferation of a wide range of measurement instruments, usually comprising several dimensions. To what extent these instruments overlap or encompass different constructs remains to be seen. As a consequence, the burnout definition applied translates into considerably different burnout prevalence estimates in the literature. Furthermore, while some researchers use a unidimensional measure of burnout, others focus on one or more dimensions. Additionally, most instruments also lack a clinically validated threshold or cutoff values for burnout diagnosis.

Future lines of research could focus on examining the relationships between self-report measures of burnout and objective biological markers (i.e., salivary cortisol) to identify which questionnaires have the highest predictive capacity for these biomarkers. In addition, adaptation and validation of the main measurement instruments to different cultural contexts is still an ongoing need.

## 7. Special Issue on “Occupational Stress and Health: Psychological Burden and Burnout”

This Special Issue includes 21 papers which bring together recent developments and studies in this field. It aims to provide a comprehensive approach to occupational health from a broad range of perspectives. The results are of use for both researchers and practitioners. Undoubtedly, the COVID-19 pandemic has impacted organizational contexts increasing the risk of stress and burnout. Burnout and stress are analyzed from different perspectives with a focus on specific occupational groups in diverse countries from several continents. Post-Traumatic Stress Disorder (PTSD) in the Military Police of Rio de Janeiro (Brazil) is investigated as well as its correlations with socio-demographic and occupational variables [131]. Gender and age differences in personal discrimination experience, burnout, and job stress among physiotherapists and occupational therapists are examined in South Korea [132]. Nurses in South Korea are further studied with respect to emotional labor, burnout, turnover intention, and medical error levels within the previous six months [133]. Healthcare workers are also the focus of another study in Japan [134], which concludes that the number of physical symptoms perceived are positively related to burnout scores. Moreover, job strain and work–family conflict are associated with an increased risk of burnout, while being married, being a parent, and job support are associated with a decreased risk of burnout. In Spain, the relationship between burnout, compassion fatigue, and psychological flexibility is analyzed in geriatric nurses [135] as well as the prevalence of emotional exhaustion, depersonalization, and possible non-psychotic psychiatric disorders in nurses during the COVID-19 pandemic [136]. In Germany [137], teachers and social workers are surveyed following a model derived from the Job Demands–Resources theory to predict effects of strains on burnout, job satisfaction, general state of health, and life satisfaction. While some professionals working in the educational sector are burned out, other develop resilience, and thus it is important to identify antecedents and profiles (e.g., support), as evidenced by another study carried out in Spain [138]. Burnout and job satisfaction are additionally examined in a sample of music therapists in Spain [139]; a higher risk of burnout is associated with working longer hours in a palliative care setting.

Although a variety of instruments have been developed and validated in different contexts, new reliable and more specific tools are timely and highly valuable to better operationalize and understand job burnout. In this line, a new scale to gauge the balance between risks and resources (*Balance*) is developed in three French-speaking countries and then longitudinally tested in several English-speaking countries [140]. Another instrument is developed to evaluate job resources and further explore the relationship between resources and psychological detachment [141]. To assess the added value of a joint use of two tools, Leclercq et al. [142] compare the diagnostic accuracy of a structured interview guide and a self-reported questionnaire, finding differences in sensitivity and specificity with implications in diagnosis and treatment. A systematic review analyses both subjective and objective measurement methods to study fatigue, sleepiness, and sleep behavior in seafarers [143]. Related to new ways to measure and study stress, the “Study on Emergency physicians’ responses Evaluated by Karasek questionnaire” (SEEK) Protocol [144] presents the design of a study protocol to examine well-being in emergency healthcare workers in order to assess and determine Karasek scores in a large sample size of emergency healthcare workers and evaluate whether there is a change in work perception (both in the short and the long term). Additionally, this protocol will allow us to explore Karasek’s associations with some biomarkers of stress and protective factors.

The identification of mediators is another promising line of research. Mérida-López et al. [145] explore in a sample of pre-service teachers in Spain the mediator role of study engagement in the relationship between self- and other-focused emotion regulation abilities and occupational commitment. A moderated-mediation model is used in China to examine the effect of perceived overqualification on emotional exhaustion, the mediating role of emotional exhaustion in the relationship between perceived overqualification and creativity, and the moderating role of pay for performance in the perceived overqualification–emotional exhaustion relationship. Occupational stressors are studied in China as mediators in the psychological capital–family satisfaction link [146]. In Brazil, the moderating role of recovery from work stress is explored in the relationship between flexibility ideals and patterns of sustainable well-being at telework [147].

Last, a growing avenue of research is devoted to leadership. Leaders’ behaviors have important consequences for both employees and organizations. In this Special Issue, ethical leadership is investigated in South Korea with respect to emotional labor and emotional exhaustion [148]. Identity leadership, team identification, and employee burnout are examined in 28 countries within the Global Identity Leadership Development (GILD) project [149]. Security-providing leadership is proposed to be a job resource to prevent employee burnout [150].

## 8. Conclusions

In this review, we have analyzed what burnout is, what are its main dimensions, what models have been proposed for the description and explanation of this syndrome, what are its antecedents and consequences, what tools allow its evaluation and how it can be intervened both at the organizational and individual level. We also present our critical vision, indicating how each specific aspect should be studied today, the future lines of research on burnout, and what the future lines of intervention in organizations should be. The most recent research published in the Special Issue on “Occupational Stress and Health: Psychological Burden and Burnout”, 21 papers, is summarized according to main areas.

There is no doubt that burnout is currently a growing concern for individuals, organizations, and society. For example, among physicians, this syndrome has reached epidemic proportions around the world, accompanied by alarming levels of depression and suicidal ideation [151]. Thus, people suffering from burnout report feeling exhausted throughout the day, and not only during their working day. In fact, just thinking about work before getting up in the morning leaves them exhausted.

Work environments with excessive work schedules and high levels of demands, as well as the need to prove that one is worthy of a certain position, leave workers emotionally drained, cynical about work, and with a low sense of personal accomplishment. Moreover, the pressure does not end with the end of the workday; new technologies, mobile devices and the lack of boundaries prevent disconnection and the necessary recovery from work.

However, burnout is not an inevitable syndrome; it can be prevented before it appears and treated during its development. Nonetheless, interventions often focus on individuals rather than organizations, even though the main causes of this syndrome are organizational factors such as work overload or role ambiguity. As Shanafelt and Noseworthy [88] point out, organizations should regularly assess the well-being of their workers, both quantitatively and qualitatively, and consider it a key performance indicator. In fact, it is likely that the relationship between burnout and job performance is underestimated because burned-out workers adopt “performance protection” strategies to maintain priority tasks and neglect low-priority secondary tasks such as, for example, dealing kindly with customers, clients, or patients [152]. In this way, evidence of the syndrome is masked until critical points are reached.

## Figures and Tables

**Figure 1 ijerph-19-01780-f001:**
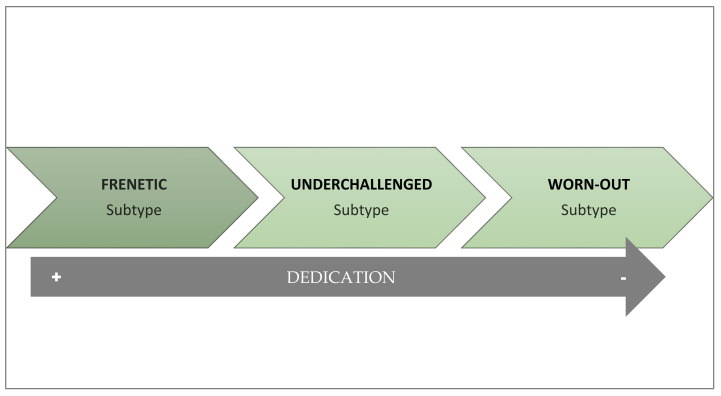
Profiles and subtypes of burnout.

**Figure 2 ijerph-19-01780-f002:**
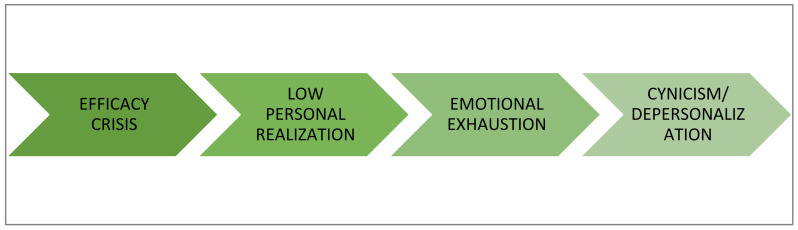
Development of burnout according to the socio-cognitive theory of the self-efficacy.

**Figure 3 ijerph-19-01780-f003:**
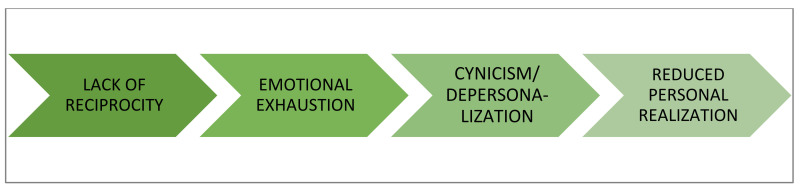
Development of burnout according to social exchange theory.

**Figure 4 ijerph-19-01780-f004:**
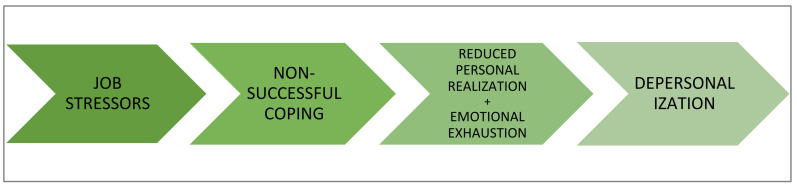
Development of burnout according to structural theory.

**Table 1 ijerph-19-01780-t001:** Burnout dimensions.

Dimension	Definition
Emotional exhaustion	This dimension manifests in the form of feelings and sensation of being exhausted by the psychological efforts made at work. It is also described in terms of weariness, tiredness, fatigue, weakening, and the subjects who manifest this type of feelings show difficulties in adapting to the work environment since they lack sufficient emotional energy to cope with work tasks.
Cynicism or depersonalization	This dimension, the interpersonal component of burnout, is defined as a response of detachment, indifference and unconcern towards the work being performed and/or the people who receive it. It translates into negative or inappropriate attitudes and behaviors, irritability, loss of idealism, and interpersonal avoidance usually towards service users, patients, and/or clients.
Reduced personal achievement	This dimension is reflected in a negative professional self-evaluation and doubts about the ability to perform the job effectively, as well as a greater tendency to evaluate results negatively. It also translates into a decrease in productivity and capabilities, low morale, as well as lower coping skills.

**Table 2 ijerph-19-01780-t002:** Summary of main demands and job resources.

Job Demands	Job Resources
Temporary pressure Interpersonal conflicts with clients and colleagues Task complexity Job insecurity Unfavorable schedule changes Qualitative and quantitative work overload Personal occupational hazards	*Individual* Technical knowledge and skills Socio-emotional skills Positive psychological capital (self-efficacy, optimism, hope and resilience) Creativity *Organizational* Time flexibility Job security Supervisor and peer support Material resources Autonomy Rewards

**Table 3 ijerph-19-01780-t003:** Individual burnout modulators.

Protectors of Burnout	Enhancers of Burnout
Agreeableness Conscientiousness Extraversion Openness to experience Positive psychological capital Problem-focused coping	Neuroticism External locus of control Type A Personality Alexithymia Emotion-focused coping

**Table 4 ijerph-19-01780-t004:** Summary of burnout interventions.

Promoted by the Organization		Promoted by the Worker	
Aimed at the Structure	Aimed at Employees	Aimed at Oneself	Aimed at Aspects of the Job
Improvement of contents and workstations	Training	Physical exercise	Time management
Humanization of work schedules and implementation of work–life balance plans	Strengths-based interventions	Mindfulness training	Job crafting
Managers’ leadership development	Coaching and guidance	Self-assessment	
Use of non-financial rewards and incentives	Creation of support groups	Psychotherapy	
Development of welcome programs			
Burnout monitoring and design of tailor-made plans			
Institutionalization of the Occupational Health and Safety Service			

**Table 5 ijerph-19-01780-t005:** Examples of training actions promoted by organizations to prevent burnout.

Actions
Self-regulation and emotional management Development of other personal resources, such as resilience, self-efficacy, hope, and optimism Conflict management Work stress management Time management Job-specific technical skills Problem solving Teamwork

**Table 6 ijerph-19-01780-t006:** Generic phases of strengths-based interventions.

1. Identification of Competencies	2. Strengths Development	3. Utilization of Strengths
They usually result in a list of the most relevant strengths. Performance appraisals and other tools such as questionnaires and strengths scales can be used for this purpose.	Organizations often set up training workshops and individual development programs in which individuals are encouraged to cultivate and refine their strengths by developing a concrete action plan.	An attempt is made to match the types of tasks to be performed with the strengths of the employees.

**Table 7 ijerph-19-01780-t007:** Types of adjustments made with job crafting.

**1. Increasing Structural Job Resources**	**2. Decreasing Job Demands**	**3. Increasing the Social Resources of Employment**	**4. Increased Demand for Challenges at Work**
Doing what is possible to develop professional skills and learn new things on the job.	Organizing work in such a way that it does not cause too much stress, is mentally less intense, as well as avoiding emotionally complicated situations with customers and colleagues and trying not to make difficult decisions at work.	Asking, if necessary, for help and feedback about the job from the supervisor and co-workers.	When an interesting project comes up, proactively offer to work on it, when there is little to do, offer help to co-workers and ask for more responsibility from the supervisor.

**Table 8 ijerph-19-01780-t008:** Instruments for assessing burnout.

**Generic Instruments**	**Specific Instruments**
Maslach Burnout Inventory (MBI) Questionnaire for the Evaluation of Burnout Syndrome at Work (CESQT) Copenhagen Burnout Inventory (CBI) Oldenburg Burnout Inventory Burnout Clinical Subtypes Questionnaire (BCSQ-36/12) Burnout Assessment Tool (BAT) Shirom–Melamed Burnout Questionnaire (SMBQ)	Maslach Burnout Inventory-Human Services Survey (MBI-HSS) Brief Burnout Questionnaire Revised for nursing staff Physician Burnout Questionnaire Teacher Burnout Questionnaire Psychologist’s Burnout Inventory Burnout Questionnaire for Athletes School Burnout Inventory Parental Burnout Inventory

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
