# Peer review of "Burnout: A Review of Theory and Measurement"

_ijerph, 2022, doi:10.3390/ijerph19031780_

Round 1

Reviewer 1 Report

This was an extensive and well-written review on burnout, which will be a valuable resource for the field. I only had a few minor wording recommendations.

  1. On page 13, line 471 it states, "leadership styles have negative effects on burnout and positive effects on their psychological resources."  Though the correlation is likely negative (i.e., more of those leadership styles leads to less burnout), this phrasing is slightly confusing (i.e., the leadership style is bad).  I might say, "leadership styles are related to decreased burnout and have positive effects on...."
  2. On page 13, line 508 it states, "Organization of psychoeducational workshops on stress and burnout that can be organized in the same organization or by outsourcing the service." It would be ideal to reduce the number of times organization/ organized is said in this sentence.

Author Response

Dear reviewer,

Thank you very much for the opportunity to revise and resubmit our manuscript #1551666 “Burnout: A Review of Theory and Measurement” to IJERPH.

We are grateful for the time and attention that you have given to our manuscript. We are also glad to hear that you believe our manuscript is an extensive, well-written review and a valuable resource for the field. Below we describe the modifications we made to the manuscript according to the comments you formulated:

R1_Comment 1: On page 13, line 471 it states, "leadership styles have negative effects on burnout and positive effects on their psychological resources."  Though the correlation is likely negative (i.e., more of those leadership styles leads to less burnout), this phrasing is slightly confusing (i.e., the leadership style is bad).  I might say, "leadership styles are related to decreased burnout and have positive effects on...."

Answer: Thank you for making us note this was unclear. This sentence has been rephrased following your suggestion: “…leadership styles are related to decreased burnout and have positive effects on employees’ psychological resources.”

R1_Comment 2: On page 13, line 508 it states, "Organization of psychoeducational workshops on stress andburnout that can be organized in the same organization or by outsourcing the service." It would be ideal to reduce the number of times organization/ organized is said in this sentence.

Answer: Thanks again you for your suggestions. This sentence has been rephrased to: “Delivery of psychoeducational workshops on stress and burnout that can be scheduled in the same organization or by outsourcing the service.”

Major changes are highlighted in blue throughout the manuscript. Several other changes have been included along the text in response to the other two reviewers. We hope you agree our changes have substantially improved our work. We look forward to hearing from you again.

Sincerely,

Dr. A. Laguia (on behalf of the authors)

Reviewer 2 Report

The article makes an adequate review of the Burnout phenomenon. The authors carry out a complete, systematic, rigorous, even pedagogical approach. However, sometimes it gives the impression that one is reading a chapter of a university book on this subject rather than a scientific article. But when trying to analyze some of the main contributions of the Special Issue on "Stress and Occupational Health: Psychological Burden and Burnout" the article ends up making sense. But the review carried out by the authors lacks a critical look. Rather, we are facing an article that develops a descriptive, analytical, well-argued and written perspective, but does not focus on the inappropriate or insufficient aspects of the approaches, perspectives, dimensions, instruments and measures that have been used to investigate the burnout phenomenon.
The authors could present their critical vision at the end of each section (four sections), indicating how each specific aspect should be studied today. It is necessary to indicate, after the exhaustive review carried out, which are, according to their criteria, the future lines of research on burnout and what the future lines of intervention in organizations should be. The latter could be done in an enumerative and synthetic way, but all these recommendations would be very pertinent to give a complete, critical and comprehensive review of burnout. This review would be publishable as a scientific article if it had this critical view on the part of the authors.

Author Response

Dear reviewer,

Thank you very much for the thorough attention that you have given to our manuscript. We are grateful for your comments, and we have included the critical vision missing in new sections 3.3, 4.5, 5.5, 6.3., entitled “Future Research”.

Additionally, your comment about the summary of main contributions of the Special Issue on "Stress and Occupational Health: Psychological Burden and Burnout" inspired us to create a specific new section (7. Special Issue on “Occupational Stress and Health: Psychological Burden and Burnout”) to emphasize the relevance of recent studies in this field. Another paper published in 2022 has been incorporated to this summary. 

Major changes are highlighted in blue throughout the manuscript. Several other changes have been included along the text in response to the other two reviewers. We hope you agree our changes have substantially improved our work. We look forward to hearing from you again.

Sincerely,

Dr. A. Laguia (on behalf of the authors)

Reviewer 3 Report

Thank you for allowing me to review this article titled: 'Burnout: A Review of Theory and Measurement.' This article presents a review which discusses the identification, description and prevention of occupational burnout.

This is a detailed, valuable review of the burnout phenomenon and would be of interest to researchers and workers across multiple disciplines and industries.

A few minor revisions required:

  • Literature presented in ‘Introduction’ is fairly dated. Can more recent literature please be included to set the current context, especially with regard to occupational burnout during the COVID pandemic.
  • Page 2 lines 45 – 50: this is a very long sentence, perhaps break it into two or three sentences with references provided for all sentences.
  • Burnout definition and development of construct is discussed in detail with alternative points of view considered.
  • Section 2.1: It is not clear if these three types of burnout flow on one from the other e.g. frenetic burnout leads to underchallenged burnout which leads to worn out burnout? Or, are they simply progressive deterioration in level of worker commitment? Figure 1 seems to imply that they lead one to the other (presence of arrows) however, I think they may be separate entities which can be represented individually on a scale of ‘Dedication’. Please clarify this in the text and in Figure 1. If they are progressive e.g. frenetic leads to underchallenged, please discuss the mechanisms of this.
  • Multiple theories of burnout each discussed briefly. Figures helpful in clarifying the mechanisms of burnout according to each theory.
  • Page 5 lines 164 – 168: this is a very long sentence, perhaps break it into two sentences with references provided for all sentences.
  • To improve flow of article, consider placing section (4) which discusses circumstances of burnout prior to section (3) which discusses consequences of burnout (reverse sections 3 & 4).
  • Consequences and triggers of burnout discussed with adequate reference to current literature.
  • Section 5.1, ‘Improving job content and jobs’, ‘Humanization of schedules’, ‘Burnout monitoring’ and ‘Institutionalization of occupational health’ sections do not contain any references. As this is a review and not empirical research, please reference where this information came from.
  • Section 5.4, ‘Time self-management’, section does not contain any references. As this is a review and not empirical research, please reference where this information came from.
  • Conclusion summarises summarizes 15 papers that comprise the Special Issue on burnout

Author Response

Dear reviewer,

We would like to thank you for your careful reading of our manuscript and guiding us along the review process. Below we describe the modifications we made to the manuscript according to the comments you formulated:

R3_Comment 1: Literature presented in ‘Introduction’ is fairly dated. Can more recent literature please be included to set the current context, especially with regard to occupational burnout during the COVID pandemic.

Answer: We have expanded the introduction including more recent references. We have included 35 new references in the manuscript (over 20 considering only the introduction).

R3_Comment 2: Page 2 lines 45 – 50: this is a very long sentence, perhaps break it into two or three sentences with references provided for all sentences.

Answer: New paragraph is as follows: “Overall, burnout syndrome is an individual response to chronic work stress that develops progressively and can eventually become chronic, causing health alterations [25]. From a psychological point of view, this syndrome causes damage at a cognitive, emotional, and attitudinal level, which translates into negative behavior towards work, peers, users, and the professional role itself [26].”

R3_Comment 3: Burnout definition and development of construct is discussed in detail with alternative points of view considered.

Thank you for recognizing our efforts.

R3_Comment 4: Section 2.1: It is not clear if these three types of burnout flow on one from the other e.g. frenetic burnout leads to underchallenged burnout which leads to worn out burnout? Or, are they simply progressive deterioration in level of worker commitment? Figure 1 seems to imply that they lead one to the other (presence of arrows) however, I think they may be separate entities which can be represented individually on a scale of ‘Dedication’. Please clarify this in the text and in Figure 1. If they are progressive e.g. frenetic leads to underchallenged, please discuss the mechanisms of this.

Answer: From this perspective, burnout may be observed as a progressively developed condition (instead of separate entities). The development of the syndrome would correspond with the burnout types as stages: in order to cope with stress and frustration, the response adopted by burnout workers consists of a progressive deterioration in levels of engagement. More references have been included to clarify this explanation.

Montero-Marin, J., Prado-Abril, J., Piva Demarzo, M. M., Gascon, S., & García-Campayo, J. (2014). Coping with stress and types of burnout: explanatory power of different coping strategies. PloS One, 9(2), e89090. https://doi.org/10.1371/journal.pone.0089090

Demarzo, M., García-Campayo, J., Martínez-Rubio, D., Pérez-Aranda, A., Miraglia, J. L., Hirayama, M. S., ... & Montero-Marín, J. (2020). Frenetic, under-challenged, and worn-out burnout subtypes among Brazilian primary care personnel: Validation of the Brazilian “burnout clinical subtype questionnaire”(BCSQ-36/BCSQ-12). International Journal of Environmental Research and Public Health, 17(3), 1081. https://doi.org/10.3390/ijerph17031081

R3_Comment 5: Multiple theories of burnout each discussed briefly. Figures helpful in clarifying the mechanisms of burnout according to each theory.

Thank you.

R3_Comment 6: Page 5 lines 164 – 168: this is a very long sentence, perhaps break it into two sentences with references provided for all sentences.

Answer: New sentence is as follows: “This approach considers that burnout is a consequence of organizational and work stressors (see section 3.1) combined with inadequate individual coping strategies [18, 19].”

R3_Comment 7: To improve flow of article, consider placing section (4) which discusses circumstances of burnout prior to section (3) which discusses consequences of burnout (reverse sections 3 & 4).

Answer: Thank you for this suggestion. Theses sections are now reversed, thus improving the flow of the article.

R3_Comment 8: Consequences and triggers of burnout discussed with adequate reference to current literature.

Thank you.

R3_Comment 9: Section 5.1, ‘Improving job content and jobs’, ‘Humanization of schedules’, ‘Burnout monitoring’ and ‘Institutionalization of occupational health’ sections do not contain any references. As this is a review and not empirical research, please reference where this information came from.

Answer: Thank you for noticing that references were missing in these sections. We have included several works.

R3_Comment 9: Section 5.4, ‘Time self-management’, section does not contain any references. As this is a review and not empirical research, please reference where this information came from.

Answer: Thank you again. We have included several references.

R3_Comment 10: Conclusion summarizes 15 papers that comprise the Special Issue on burnout

Answer: Following also comments written by Reviewer 2, we have created a specific new section (7. Special Issue on “Occupational Stress and Health: Psychological Burden and Burnout”) to emphasize the relevance of recent studies in this field summarizing the now 21 papers that comprise the Special Issue on burnout. Thus, the new section 8. Conclusions only reflects the conclusions. 

Major changes are highlighted in blue throughout the manuscript. Several other changes have been included along the text in response to the other two reviewers. We have made all efforts to convince you of the quality of our study and the contribution it makes to the literature. We look forward to hearing from you again.

Sincerely,

Dr. A. Laguia (on behalf of the authors)